

# What can we learn about tropospheric OH from satellite observations of methane?

Elise Penn[1], Daniel J. Jacob[2], Zichong Chen[2], James D. East[2], Melissa P. Sulprizio[2], Lori Bruhwiler[3], Joannes D. Maasakkers[4], Hannah Nesser[5], Zhen Qu[6], Yuzhong Zhang[7], and John Worden[5]

[1]Department of Earth and Planetary Sciences, Harvard University, Cambridge, MA, USA,
[2]Harvard John A. Paulson School of Engineering and Applied Sciences, Harvard University, Cambridge, MA, USA,
[3]NOAA Earth System Research Laboratory, Global Monitoring Division, Boulder, CO, USA,
[4]SRON Netherlands Institute for Space Research, Leiden, the Netherlands
[5]Jet Propulsion Laboratory, California Institute of Technology, Pasadena, CA, USA,
[6]Department of Marine, Earth, and Atmospheric Sciences, North Carolina State University, Raleigh, NC, USA
[7]Key Laboratory of Coastal Environment and Resources of Zhejiang Province (KLaCER), School of Engineering, Westlake University, Hangzhou, Zhejiang, China

*Correspondence to*: Elise Penn (epenn@g.harvard.edu)

**Abstract.** The hydroxyl radical (OH) is the main oxidant in the troposphere and controls the lifetime of

many atmospheric pollutants including methane. Global annual mean tropospheric OH concentrations ($[\overline{OH}]$) have been inferred since the late 1970s using the methyl chloroform (MCF) proxy. However, concentrations of MCF are now approaching the detection limit, and a replacement proxy is urgently needed. Previous inversions of GOSAT satellite measurements of methane in the shortwave infrared (SWIR) have shown success in quantifying $[\overline{OH}]$ independently of methane emissions, and observing system simulations have suggested that thermal infrared (TIR) measurements may

provide additional constraints on OH. Here we combine TIR satellite observations of methane from AIRS with SWIR observations from GOSAT in a three-year (2013-2015) analytical Bayesian inversion optimizing both methane emissions and OH concentrations. We examine how much information can be achieved on the interannual, seasonal, and latitudinal features of the OH distribution using information from MCF data as well as the ACCMIP ensemble of global atmospheric chemistry models to construct a full prior error covariance matrix for OH concentrations for use in the inversion. This is

essential to avoid overfit to observations. Our results show that GOSAT alone is sufficient to quantify $[\overline{OH}]$ and its interannual variability independently of methane emissions, and that AIRS adds little information. The ability to constrain the latitudinal variability of OH is limited by strong error correlations. There is no information on OH at mid-latitudes, but there is some information on the NH/SH interhemispheric ratio, showing this ratio to be lower than currently simulated in models. There is also some information on the seasonal variation of OH concentrations, though it mainly confirms that

simulated by models. Future satellite observations of methane will continue to improve our understanding of methane emissions and consequently $[\overline{OH}]$ and its interannual variability.



## 1 Introduction

The hydroxyl radical (OH) is the main oxidant in the troposphere. It determines the lifetimes of most atmospheric species
removed by oxidation such as methane (a major greenhouse gas), non-methane volatile organic compounds (NMVOCs,
important for air quality), and hydrogenated halocarbons (contributing to stratospheric ozone loss). The global OH
concentration and its trend have been monitored indirectly since the 1980s by measuring the concentration of
methylchloroform (MCF), an industrial solvent removed from the atmosphere by reaction with OH (Bousquet et al., 2005;
Krol et al., 1998; Lovelock, 1977; Patra et al., 2020; Prinn et al., 1987). MCF was banned in the 1990s because of its
contribution to stratospheric ozone depletion, and its concentration is now approaching the detection limit where it loses its
value as a proxy for OH (Liang et al., 2017). An observation system simulation experiment (OSSE) previously suggested
that a combination of thermal infrared (TIR) and shortwave infrared (SWIR) satellite observations of atmospheric methane
could provide a continued proxy for global OH going forward (Zhang et al., 2018). Here we evaluate this idea with a joint
inversion of AIRS and GOSAT satellite measurements for 2013-2015, examining the capability of the observations to
quantify global OH concentrations as well as interannual, seasonal, and latitudinal variations.

The OH concentration is controlled by complex photochemistry (Lelieveld et al., 2016; Levy, 1971; Logan et al., 1981). The
primary source is UV-B photolysis of ozone in the presence of water vapor. The main sinks are reactions with carbon
monoxide (CO), methane, and NMVOCs, resulting in a lifetime ~1 second, and producing peroxy radicals that can be
recycled to OH by reaction with nitric oxide (NO). The global mean tropospheric OH concentration is commonly expressed
as the lifetime of methane against oxidation by tropospheric OH, $\tau_{CH4}^{OH}$. From the methylchloroform proxy one infers the
tropospheric lifetime of OH $\tau_{CH4}^{OH} = 10.2_{-0.7}^{+0.9}$ years for 2000 (Prinn et al., 2005). Current atmospheric chemistry models find
a methane lifetime of 8.4 ± 0.3 years, implying that OH in the models is too high (Stevenson et al., 2020).

Although models are generally consistent in their simulations of global mean OH concentrations, there are large
disagreements in the regional distributions of OH concentrations driven by NO$_x$ and NMVOC distributions (Naik et al.,
2013; Zhao et al., 2020), chemical mechanisms (Murray et al., 2021), clouds (Liu et al., 2006; Voulgarakis et al., 2009), and
other meteorological variables (He et al., 2021). Models consistently simulate higher OH in the Northern Hemisphere (NH)
than the Southern Hemisphere (SH) (Naik et al., 2013; Stevenson et al., 2020). MCF observations, by contrast, suggest no
interhemispheric gradient (Patra et al., 2014), or slightly higher OH in the SH (Montzka et al., 2000). Models may have
excessive OH in the northern hemisphere because of underestimated CO (Naik et al., 2013).

Understanding year-to-year variability and decadal-scale trends in OH concentrations is important for attributing the cause of
methane fluctuations (Turner et al., 2017), including the recent acceleration of the methane trend (Laughner et al., 2021; Qu
et al., 2022; Stevenson et al., 2022). Methane is emitted from a range of poorly quantified sources including wetlands,



livestock, waste, fuel exploitation, rice paddies, and open fires (Saunois et al., 2020). These sources could be responsible for methane interannual variability and trends but so could OH concentrations (Turner et al., 2017). The El Nino Southern Oscillation (ENSO) drives interannual variability in model OH due to its influence on lightning (Anderson et al., 2021; Murray et al., 2013; Turner et al., 2018), water vapor (Anderson et al., 2021; Turner et al., 2018), and CO emitted from

70 biomass burning (Zhao et al., 2020). Models and measurements show a 5% range of interannual variability of OH over the last 30 years though with no temporal correlation between the two (Szopa et al., 2021). Models find increasing OH from 1980 to present driven by increases in anthropogenic $NO_x$ emissions (Gaubert et al., 2017; Naik et al., 2013; Stevenson et al., 2020; Zhao et al., 2019). By contrast, MCF observations indicate OH increasing from 1980 to 2005 but then flat or decreasing after 2005 (Nicely et al., 2018; Rigby et al., 2017; Stevenson et al., 2020; Turner et al., 2017).

Many studies have used satellite observations of methane to infer methane emissions using specified OH concentrations to optimize methane sources (Turner et al., 2015), while others have attempted to optimize both methane sources and OH concentrations by exploiting differences in spatial/seasonal impacts on methane concentrations (Maasakkers et al., 2016; Zhang et al., 2021) or by including in the inversion complementary information from observations of MCF (Cressot et al.,

2014; Cressot et al., 2016) or formaldehyde and CO (Yin et al., 2021). Inversions of GOSAT (SWIR) satellite observations of methane alone can constrain global mean OH about as well as MCF and infer a flat interhemispheric gradient, although posterior errors may be too optimistic (Lu et al., 2021; Maasakkers et al., 2019; Zhang et al., 2021). Zhang et al., (2018) proposed that TIR satellite observations of methane, which have sensitivity to the free troposphere and broader coverage over oceans and at night, may reduce error correlation between OH and methane emissions.

Satellite-based observations of methane in the TIR have been made continuously since 2002 by several instruments: AIRS (2002-present), TES (2004-2011), IASI (2007-present), CrIS (2011-present), and GOSAT-2 (2018-present) (Jacob et al., 2016). TIR observations have received little attention in inverse studies because they are not sensitive to methane near the surface (Wecht et al., 2012). Direct applications of TIR satellite observations have mostly focused on processes affecting the

90 free troposphere, such as detecting stratospheric intrusions (Xiong et al., 2013), methane emissions from large wildfires (Ribeiro et al., 2018; Xiong et al., 2010), interannual variations in mid-troposphere methane in response to ENSO (Corbett et al., 2017), seasonal fluctuations of methane in response to fossil fuel and rice paddy emissions in China (X. Zhang et al., 2011), and differences of seasonality compared to surface observations (Zhou et al., 2023). The combination of SWIR and TIR observations has been used to develop lower troposphere methane products including with GOSAT+AIRS (Worden et

al., 2015), GOSAT+IASI (Schneider et al., 2022), and GOSAT-2 (Kuze et al., 2022; Suto, 2022).

Here we combine TIR observations from AIRS with SWIR observations from GOSAT in a three-year 2013-2015 inversion optimizing both methane emissions and OH concentrations. We use an analytical solution that provides formal characterization of posterior error statistics (including error correlations) and information content as part of the inversion.





We place particular focus on the ability of the inversion to quantify global mean OH concentrations, interannual variability, and latitudinal and seasonal variations. This involves careful characterization of prior error covariances using OH concentrations from the ACCMIP model ensemble (Naik et al., 2013).

## 2 Data and Methods

We use 3 years (2013-2015) of satellite observations from GOSAT and AIRS (Sect. 2.1), to optimize a state vector of OH

distributions and annual methane emissions. The observations are assembled in an observation vector $y$ with total dimension $m$. The state vector $x$ comprises $n$ elements describing annual gridded non-wetland methane emissions, monthly subcontinental wetland methane emissions, and mean OH concentrations for individual years in different latitudinal bands and seasons (Sect. 2.2). Optimization is done by Bayesian inference using a prior estimate $x_A$ for the state vector and error covariances for that prior estimate ($S_A$) and for the observations ($S_o$) (Sect. 2.3), together with the GEOS-Chem chemical

transport model $y = F(x)$ expressing the sensitivity of the observations to the state vector (Sect. 2.4). We use an analytical solution for minimization of the Bayesian cost function $J(x)$ to yield the optimal value (posterior estimate) $\hat{x}$ of the state vector, the posterior error covariance matrix $\hat{S}$, and metrics of information content (Sect. 2.5). The subsections below describe these different elements of the inversion except for the prior error covariance matrix of OH concentrations, which will be presented in a dedicated Sect. 3. Throughout this paper, we refer to "OH concentrations" or [OH] for a given domain

as the mass-weighted average OH number density for that domain, and the global annual mean OH concentrations as $[\overline{OH}]$.

### 2.1 Satellite data

GOSAT (Greenhouse gases Observing SATellite), launched in 2009, detects methane by solar backscatter in the SWIR using the TANSO-FTS (Thermal And Near infrared Sensor for carbon Observation - Fourier Transform Spectrometer) instrument. In its default operating mode, GOSAT provides 10.5 km-diameter nadir observations of radiance separated by about 250 km

along-track and cross-track on a sun-synchronous orbit with an equatorial overpass at about 1300 local solar time (LST). We use the University of Leicester $CO_2$-proxy methane retrieval v9.0 (Parker and Boesch, 2020), which uses the GOSAT observations in the 1.65 µm band to retrieve methane as a column-averaged dry air mixing ratio $X_{CH4}$ with a vertical sensitivity profile (column averaging kernel) of near-unity in the troposphere.

AIRS (Atmospheric Infrared Sounder), launched in 2002, detects methane by observing TIR radiation emitted by the Earth. AIRS provides 15 km-diameter nadir observations across a 1250 km swath with equatorial overpasses at about 0130 and 1330 LST, resulting in global coverage twice per day. We use the optimal estimation MUSES-AIRS retrieval of methane in the 8 and 12 µm bands, which provides 26-level profiles of dry-air methane mixing ratio (Kulawik et al., 2021). The AIRS instrument has less than two degrees of freedom for signal per measurement and little sensitivity to the lower troposphere.

We therefore convert the vertical profiles to a column-averaged dry air mixing ratio $X_{CH4}$ above 600 hPa, with column





averaging kernels featuring maximum sensitivity to the upper troposphere. See Worden et al., (2015) for typical GOSAT and AIRS column averaging kernels.

For both AIRS and GOSAT, we remove measurements flagged for low quality, negative values, and surface pressures
differing by more than 50 hPa from the local GEOS-Chem surface pressure which would indicate unresolved topography. We do not use GOSAT sunglint measurements because of their sparsity and seasonal sampling bias (Maasakkers et al., 2019). We also exclude measurements poleward of 60° due to model stratospheric bias in interpreting methane column observations in the polar vortex (Stanevich et al., 2020; Turner et al., 2015; Zhang et al., 2021). We include both daytime and nighttime measurements for AIRS, as we find no significant biases between them. This results in 600,000 successful
retrievals for GOSAT and 2.5 million for AIRS.

In order to compare satellite retrievals to the GEOS-Chem simulations, we produce a model column sampled in the same manner as the satellite data. For each AIRS and GOSAT observation, we select the coincident GEOS-Chem grid cell and interpolate the GEOS-Chem methane mixing ratio profile, which is on 47 vertical levels, to the AIRS profile (26 vertical
levels) and the GOSAT profile (20 vertical levels) using a mass-conserving interpolation algorithm described in Keppens et al. (2019) with Python code available on GitHub at https://github.com/pennelise/GOOPy (Penn and Nesser, 2024). We call these interpolated profiles $c_m$. We then translate these profiles to column-averaged dry air mixing ratios using the column averaging kernel $a$. The column averaging kernel is based on mixing ratio and does not include different pressure weights for each level (Boesch et al., 2011), so we apply the pressure weighting function ($h$) provided in the GOSAT and AIRS data
products. For an individual satellite $X_{CH4}$ observation $y$, we derive the corresponding model value $y_m$ using:

$$y_m = h^T((I - a)^T c_a + a^T c_m) \tag{1}$$

where $I$ is the unit vector and $c_a$ is the prior profile provided by the GOSAT and AIRS products, which come from the MACC-II methane inversion and TOMCAT stratospheric chemistry model for GOSAT and from the MOZART atmospheric
chemistry model for AIRS.

Figure 1 shows satellite observations from 2013 for GOSAT and AIRS compared to a 2013 GEOS-Chem simulation driven by GOSAT-optimized emissions from Lu et al. (2021). As expected, GOSAT is globally unbiased relative to this GEOS-Chem simulation (-2 ± 12 ppb), but AIRS is biased low (-19 ppb ± 24 ppb), and so we apply a correction of +19 ppb to the
AIRS data to ensure consistency with GOSAT.



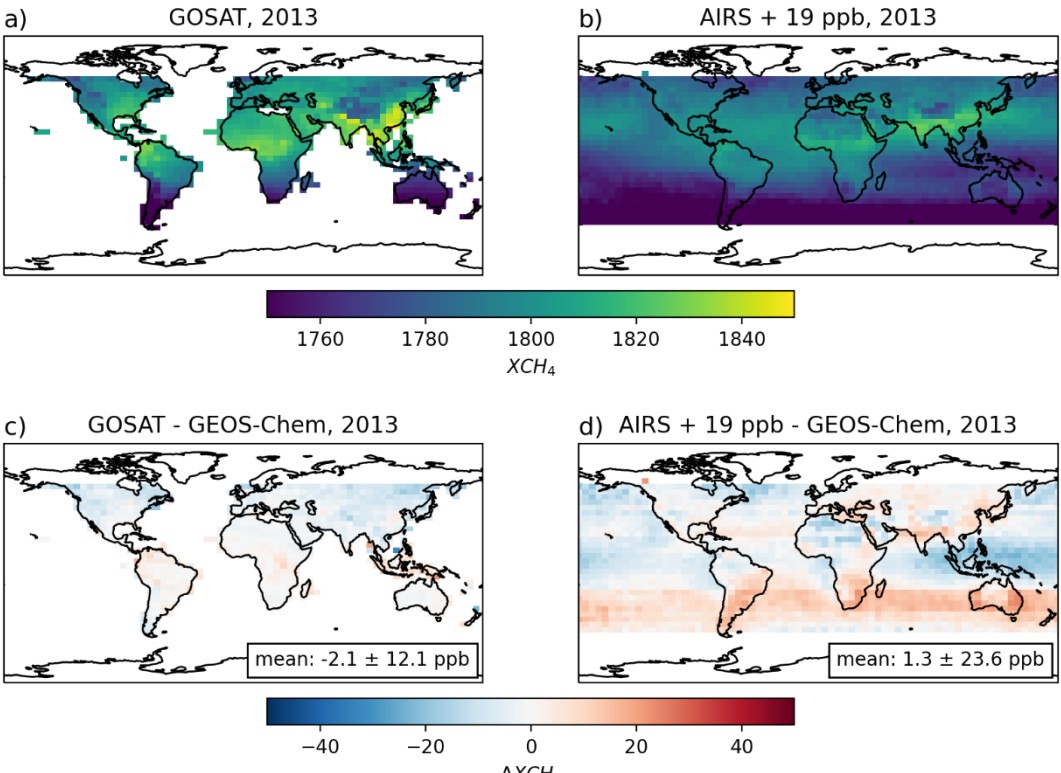

**Figure 1: GOSAT and AIRS observations of annual mean methane dry column mixing ratio ($X_{CH4}$) in 2013, binned by 4°x5° grid cells. GOSAT sunglint and observations poleward of 60° are not included. The bottom panels compare these observations with a GEOS-Chem simulation driven by 2013 posterior emissions from an inversion of GOSAT observations (Lu et al., 2021). A +19 ppb global bias correction is applied to AIRS on the basis of this comparison. Means and standard deviations of the differences between the satellite observations and GEOS-Chem are given inset.**

## 2.2 State vector and prior estimates

We optimize a state vector including annual gridded non-wetland emissions, monthly subcontinental wetland emissions, and OH distributions. Non-wetland emissions consist of 1009 4°x5° grid cells over land for each year (1009x3 = 3027 elements). Wetland emissions are optimized for each month and in 14 subcontinental regions following Bloom et al. (2017) (12x14x3 = 504 elements). OH concentrations are optimized for each season and year in four latitude bands of 30° each from 60°S to 60°N (4x4x3 = 48 elements). This results in $n$ = 3579 total state vector elements.

We define $\boldsymbol{K} = \partial\boldsymbol{y}/\partial\boldsymbol{x}$ as the $m \times n$ Jacobian matrix describing the dependence of satellite observations on the state vector as simulated by GEOS-Chem. We calculate the Jacobian by perturbing each element of the state vector by 50% (for emissions) and 20% (for [OH]), resulting in $n + 1$ = 3580 forward model runs. The forward model is strictly linear in the





relationship of concentrations to emissions, and the assumption of linearity is also acceptable for the relationship to OH concentrations in a 3-year simulation. Thus $K$ fully defines GEOS-Chem for the purpose of the inversion.

The state vector elements are optimized in the inversion as scaling factors relative to prior estimates. We use the same prior estimates as Lu et al. (2021). Default prior anthropogenic emissions are from the EDGAR inventory v4.3.2 (Crippa et al., 2018) and are superseded for the US by the gridded EPA inventory of Maasakkers et al. (2016) and globally for oil, gas, and coal by the GFEI inventory of Scarpelli et al. (2020). Prior anthropogenic emissions are assumed constant except for manure

and rice for which we apply seasonal scaling factors (Maasakkers et al., 2016; Zhang et al., 2016). Prior wetland emissions are from WetCHARTS v1.0 with 0.5°x0.5° spatial resolution and monthly temporal resolution, and including the partitioning into 14 subcontinental regions for use in inversions (Bloom et al., 2017). Additional prior emissions include the GFED inventory for fires at daily resolution (Randerson et al., 2017), and geologic sources from Etiope et al. (2019) scaled to the global total from Hmiel et al. (2020). Prior tropospheric OH concentrations (Figure 2) are archived monthly mean values

from an older (version 5) GEOS-Chem simulation on the 4°x5° grid (Wecht et al., 2014). The mass-weighted annual mean tropospheric OH concentration is $[\overline{OH}] = 11.2 \times 10^5$ molec. cm$^{-3}$ consistent with the MCF-derived estimate from 2000 of $[\overline{OH}] = 10.8^{+0.77}_{-0.85} \times 10^5$ molec. cm$^{-3}$ (Prinn et al., 2005). More recent versions of GEOS-Chem overestimate $[\overline{OH}]$ (Shah et al., 2023), consistent with the current generation of models (Stevenson et al., 2020).

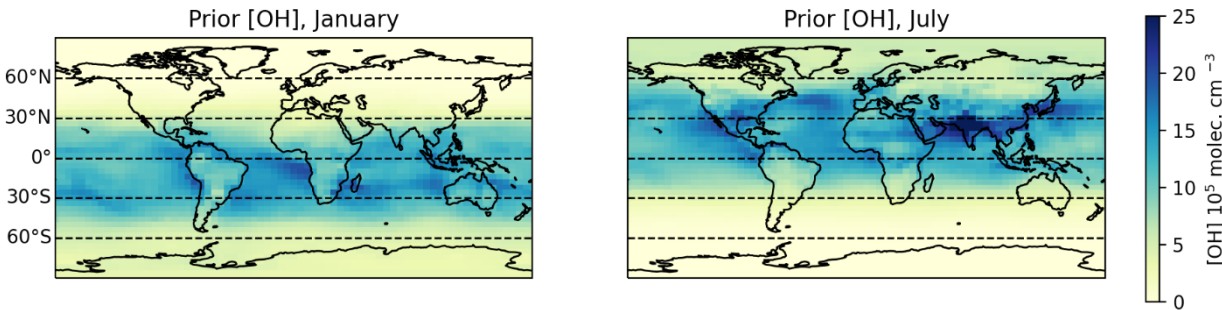

**Figure 2. Mass-weighted tropospheric OH concentrations in GEOS-Chem used as prior estimates for the inversions. Monthly mean values for January and July are shown.**

## 2.3 Error estimates

The inversion requires specification of both observing system and prior error covariance matrices. The observing system error includes contributions from the measurement and from the forward model. We use the residual error method described in Heald et al. (2004) to derive it. We first split the observations into monthly 4°x5° grid cell subsets and compare observations within each subset to the GEOS-Chem simulation $F(x)$ using prior values. We then assume that the model bias





$(b = \overline{F(x_A) - y})$ within each subset is due to error on the prior estimates, and that the residual represents the observing

system error. We find in this manner mean observing system error standard deviations of 12 ppb for GOSAT and 22 ppb for

AIRS, mostly contributed by the retrieval error with reported error standard deviations averaging 10 ppb for GOSAT and 16

ppb for AIRS. Our observing system error standard deviation for GOSAT is consistent with previous estimates (e.g. Lu et

al., 2021; Qu et al., 2021; Zhang et al., 2021). We construct the observing system error covariance matrix assuming no error

correlation between individual observations (diagonal matrix).

Prior error standard deviations for non-wetland emissions are assumed to be 50% of emissions for each 4°x5° grid cell with

no error covariance between grid cells. For wetland emissions, we calculate the full prior error covariance matrix between all

14 regions and 36 months from the WetCHARTs model ensemble following Bloom et al. (2017), and then shrink the off-

diagonal terms following Schäfer and Strimmer (2005) to ensure that the matrix is positive definite. Prior error estimates for

the OH elements of the state vector are derived in Section 3.

## 2.4 Forward Model

We use the GEOS-Chem version 12.7.1 $CH_4$ simulation (DOI: 10.5281/zenodo.3676008) on a 4°x5° grid with 47 vertical

layers as forward model for the inversion. Atmospheric transport is driven by the Modern-Era Retrospective Analysis,

version 2 (MERRA-2) assimilated meteorological fields for 2013-2015 from the NASA Global Modeling and Assimilation

Office. In addition to the tropospheric OH fields optimized in the inversion (Section 2.2), minor methane sinks in GEOS-

Chem include stratospheric loss prescribed with 2-D oxidant fields (Murray et al., 2013), oxidation by tropospheric Cl

following Wang et al. (2019), and soil uptake from the MeMo inventory (Murguia-Flores et al., 2018). Initial conditions for

January 1, 2013 come from the GOSAT-optimized posterior simulation of Lu et al. (2021) and are globally unbiased with

respect to GOSAT and adjusted AIRS observations as described in Section 2.1.

**2.5 Inversion**

We perform three inversions: "GOSAT-only" optimized with GOSAT observations, "AIRS-only" optimized with AIRS

observations, and "GOSAT+AIRS" optimized with both. The equations below are for the inversion using both GOSAT and

AIRS observations. Because we assume no error correlations between the instruments, an inversion with only one instrument

can be derived by removing all terms pertaining to the other instrument.

We minimize a Bayesian cost function that accounts for the distance from the prior estimate ($x_A$) and the satellite

observations ($y$), weighted by the inverse of the prior ($S_A$) and observing system ($S_O$) error covariance matrices, and

including an additional regularization factor ($\gamma$). Observing system components from GOSAT and AIRS are denoted by

subscripts. Assuming normal errors, and further assuming no correlation between GOSAT and AIRS errors, the cost function

is given by:





$$J(x) = (x - x_A)^T S_A^{-1}(x - x_A)$$
$$+\gamma_{\text{GOSAT}}(y_{GOSAT} - K_{GOSAT}x)^T S_{O,GOSAT}^{-1}(y_{GOSAT} - K_{GOSAT}x) \tag{2}$$
$$+\gamma_{\text{AIRS}}(y_{AIRS} - K_{AIRS}x)^T S_{O,AIRS}^{-1}(y_{AIRS} - K_{AIRS}x)$$

We can then solve $\min(J(x))$ analytically by setting $\partial J/\partial x = 0$ and obtain the posterior solution $\hat{x}$ (Rodgers, 2000):

$$\hat{x} = x_A + G_{GOSAT}(y_{GOSAT} - K_{GOSAT}x_A) + G_{AIRS}(y_{AIRS} - K_{AIRS}x_A) \tag{3}$$

where $\hat{x}$ is the posterior estimate for the state vector and $G_{AIRS}$ and $G_{GOSAT}$ are the gain matrices:

$$G_{AIRS} = S_A K_{AIRS}^T \left( K_{AIRS} S_A K_{AIRS}^T + \frac{S_{O,AIRS}}{\gamma_{\text{AIRS}}} \right)^{-1} \tag{4}$$

$$G_{GOSAT} = S_A K_{GOSAT}^T \left( K_{GOSAT} S_A K_{GOSAT}^T + \frac{S_{O,GOSAT}}{\gamma_{\text{GOSAT}}} \right)^{-1}$$

The analytical solution also yields a closed-form expression for the posterior error covariance matrix $\hat{S}$ characterizing the normal error on $\hat{x}$ :

$$\hat{S} = \left( \gamma_{\text{GOSAT}} K_{GOSAT}^T S_{O,GOSAT}^{-1} K_{GOSAT} + \gamma_{\text{AIRS}} K_{AIRS}^T S_{O,AIRS}^{-1} K_{AIRS} + S_A^{-1} \right)^{-1} \tag{5}$$

We can also derive the averaging kernel matrix $\partial \hat{x}/\partial x$ that describes the sensitivity of the posterior estimate to the true state:

$$A = I_n - \hat{S}S_A^{-1} \tag{6}$$

The trace of the averaging kernel gives us the Degrees of Freedom for Signal (DOFS), which describes the number of pieces
of independent information derived from the inversion.

For some of our applications we will aggregate state vector elements into a reduced state vector $x_{red}$ using a summation matrix $W$:





$$\widehat{x}_{red} = W\widehat{x} \tag{7}$$

and derive the corresponding averaging kernel ($A_{red}$) and posterior error covariance ($\widehat{S}_{red}$) for the aggregated solution:

$$A_{red} = WAW^* \tag{8}$$

$$\widehat{S}_{red} = W\widehat{S}W^T \tag{9}$$

where $W^*$ is the Moore-Penrose pseudoinverse of $W$.

The regularization factor $\gamma$ is intended to avoid overfitting to observations caused by not accounting for error covariance in the observing system (matrix $S_O$) We determine the appropriate value for $\gamma$ using the technique described in Lu et al. (2021). The sum of prior terms in the posterior value of the cost function, $J_A(\widehat{x}) = (\widehat{x} - x_A)^T S_A^{-1}(\widehat{x} - x_A)$, should follow a chi-

square distribution with expected value $J_A(\widehat{x}) = n$, and we adjust $\gamma$ to achieve this. We determine $\gamma_{GOSAT}$ and $\gamma_{AIRS}$ separately using GOSAT-only and AIRS-only inversions. We find in this manner $\gamma_{GOSAT} = 0.2$ and $\gamma_{AIRS} = 0.1$. To provide equal weight to [OH] and methane emissions in the cost function, we follow Maasakkers et al. (2019) and scale the OH prior error covariance matrix $S_{A,OH}$ by the ratio of the number of emission state vector elements to OH state vector elements, or 3531/48, before inserting them into the full prior error matrix $S_A$.



### 3 Construction of prior error covariance matrix for OH concentrations

GOSAT observations of methane have been used in inversions to infer the global mean tropospheric OH concentration, its interannual variability, and its interhemispheric difference (Maasakkers et al., 2019; Qu et al., 2021, 2024; Zhang et al., 2021). Here we explore how much information satellite observations can actually provide on OH concentrations by including in the state vector the OH concentrations in individual years (2013-2015), four latitudinal bands, and four seasons, for a total of 48 state vector elements (Section 2.2) for which we can diagnose posterior error correlations and information content. This requires accounting for prior error correlations between these different elements, as represented in a 48×48 matrix $S_{A,OH}$.

We construct the prior error covariance matrix for OH in the following manner. First, we specify the error statistics for global annual mean mass-weighted tropospheric OH concentrations, $[\overline{OH}]$. This includes a systematic error of 10% within the MCF constraint (Prinn et al., 2005) and an interannual variability error that we estimate to be 5% on the basis of interannual variability of model and MCF-derived $[\overline{OH}]$ reported by Holmes et al. (2013). Thus the prior error covariance matrix for $[\overline{OH}]$ in our three simulation years (2013-2015), in unit of fractional error variances and covariances, is given by a 3×3 matrix $\overline{S_{A,OH}} = (\sigma_{ij}^2)$:

$$\overline{S_{A,OH}} = \begin{bmatrix} 0.05^2 + 0.1^2 & 0.1^2 & 0.1^2 \\ 0.1^2 & 0.05^2 + 0.1^2 & 0.1^2 \\ 0.1^2 & 0.1^2 & 0.05^2 + 0.1^2 \end{bmatrix} \tag{10}$$

Prior error correlations between OH concentrations in different latitudinal bands and seasons should account for our current knowledge of the OH distribution. We use for this purpose monthly mean output for one year from the ensemble of 11 independent ACCMIP global atmospheric chemistry models reported in Naik et al. (2013). All ACCMIP models include the same anthropogenic emissions of $NO_x$, CO, and NMVOCs. They have different natural emissions, chemical mechanisms, and meteorology. Global distributions of OH concentrations in each ACCMIP model were presented previously in Zhang et al. (2018). For each ACCMIP model, we calculate the mass-weighted integral of OH concentrations vertically up to 200 hPa for each 30° latitude band for each season. We then compute the variances and covariances between each latitude band and season across the ensemble of ACCMIP models. The resulting 16x16 covariance matrix for the ACCMIP models $S_{A,AM}$ is taken as the error covariance matrix in the spatial-seasonal distribution of OH for the inversion, with error standard deviations represented by a diagonal matrix $D$.

Figure 3 shows the spatial and seasonal error correlation matrix $R_{A,AM}$ and the error standard deviations $D$ calculated directly from the ACCMIP ensemble, such that $S_{A,AM} = D R_{A,AM} D$. We find strong error correlations in the tropics for all seasons,



indicating a commonality of effects driving [OH] differences between models. Error correlations are also strong between mid-latitudes summer and the tropics, likely for the same reasons. Mid-latitude OH concentrations in other seasons show much weaker error correlations, implying that they are driven by different photochemistry and emissions as might be expected. Northern and southern midlatitudes are highly correlated in their respective winters.

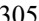

**Figure 3: Error correlations for model OH concentrations in different latitude bands and seasons (denoted $R_{A,AM}$ in the text). Pearson's error correlation coefficients are calculated for the ensemble of 11 different ACCMIP models. The mean and standard deviation of the ACCMIP ensemble for each latitude and season is inset above.**



We replicate the 16×16 spatial-seasonal OH error covariance matrix $\boldsymbol{S_{A,AM}}$ constructed from the ACCMIP data to create a 48×48 error covariance matrix for the three years of our analysis, resulting in the block matrix:

$$
\begin{bmatrix}
S_{A,AM} & S_{A,AM} & S_{A,AM} \\
S_{A,AM} & S_{A,AM} & S_{A,AM} \\
S_{A,AM} & S_{A,AM} & S_{A,AM}
\end{bmatrix}
\tag{11}
$$

This matrix is low rank because it was constructed with information from only 11 models to estimate 48 state vector elements. We use the method of Schäfer and Strimmer (2005) to shrink the off-diagonal errors and produce a matrix that is positive definite and invertible. Schäfer and Strimmer (2005) show that their method produces a more accurate estimate of the true error covariance matrix (where accuracy is defined by comparison of the true and estimated eigenvalues). After off-
diagonal shrinkage, matrices along the diagonal of the block matrix differ from those off-diagonal. We refer to the resulting 16x16 covariance matrices of spatial-seasonal errors within years as $\boldsymbol{S_{A,AM}}''$, and between years as $\boldsymbol{S_{A,AM}}'$. Additionally, we refer to the error variances of the global mean $[\overline{OH}]$ for one year inferred from these matrices as $\sigma_{AM}^2{}''$ and $\sigma_{AM}^2{}'$.

We can then construct $\boldsymbol{S_{A,OH}}$ from the regularized ACCMIP covariance matrices $\boldsymbol{S_{A,AM}}''$ and $\boldsymbol{S_{A,AM}}'$ scaled by the annual
mean error variances inferred from the MCF observations $\sigma_{ij}^2$ (Eq. (10)) and the spatial-seasonal error variances inferred from the ACCMIP model $\sigma_{AM}^2{}''$ and $\sigma_{AM}^2{}'$. We can formulate $\boldsymbol{S_{A,OH}}$ as a block matrix, where each block is an appropriately scaled ACCMIP covariance matrix for one year, as shown:

$$
\boldsymbol{S_{A,OH}} =
\begin{bmatrix}
\dfrac{\sigma^2{}_{11}}{\sigma_{AM}^2{}''} \boldsymbol{S_{A,AM}}'' & \dfrac{\sigma^2{}_{12}}{\sigma_{AM}^2{}'} \boldsymbol{S_{A,AM}}' & \dfrac{\sigma^2{}_{13}}{\sigma_{AM}^2{}'} \boldsymbol{S_{A,AM}}' \\
\dfrac{\sigma^2{}_{21}}{\sigma_{AM}^2{}'} \boldsymbol{S_{A,AM}}' & \dfrac{\sigma^2{}_{22}}{\sigma_{AM}^2{}''} \boldsymbol{S_{A,AM}}'' & \dfrac{\sigma^2{}_{23}}{\sigma_{AM}^2{}'} \boldsymbol{S_{A,AM}}' \\
\dfrac{\sigma^2{}_{31}}{\sigma_{AM}^2{}'} \boldsymbol{S_{A,AM}}' & \dfrac{\sigma^2{}_{32}}{\sigma_{AM}^2{}'} \boldsymbol{S_{A,AM}}' & \dfrac{\sigma^2{}_{33}}{\sigma_{AM}^2{}''} \boldsymbol{S_{A,AM}}''
\end{bmatrix}
\tag{12}
$$

This enforces error variances and covariances for annual global mean OH concentrations identical to the values $\sigma_{ij}^2$ from Eq. (10).

We refer to Eq. (12) as the full-correlations error covariance matrix. We will also test the effect of simpler OH correlation assumptions on inversion results, while keeping the state vector the same. First is a no-correlations error covariance matrix
that assumes diagonal errors for the OH concentration, with no error correlation between years, seasons, or latitude bands. Second is a correlated years error covariance matrix that includes error correlations between years but with no spatial-



seasonal structure. We scale the correlated-years error covariance matrix such that the error (co)variances for $\overline{[\text{OH}]}$ are identical to $\overline{S_{A,OH}}$ in Eq. (10). We cannot do the same for the no-correlations error covariance matrix because it is diagonal; however we scale it such that the error variance of the three-year average is identical to that represented by $\overline{S_{A,OH}}$. The

variance of the three-year average is therefore identical for all three error covariance matrices.

# 4 Results & discussion

## 4.1 Quantifying emissions

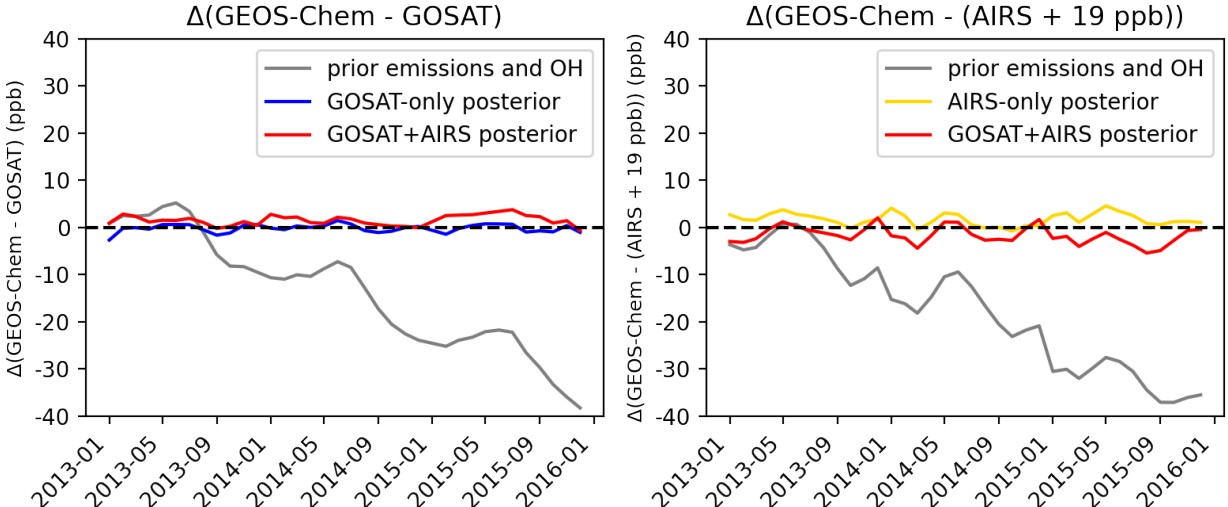

**Figure 4. Difference between the global mean dry column mixing ratio ($X_{\text{CH4}}$) simulated by GEOS-Chem and observed by GOSAT**
**(left) and AIRS (right). Monthly mean results are shown for the 2013-2015 inversion period. The GEOS-Chem simulation is driven by either prior or posterior values for emissions and OH concentrations. Posterior values are from inversions using either GOSAT or AIRS observations or both. The 19 ppb correction applied to AIRS observations is to remove the bias with GOSAT (Sect 2.1).**

Figure 4 compares the global mean dry column mixing ratio ($X_{\text{CH4}}$) simulated by GEOS-Chem and observed by GOSAT and
AIRS. The prior simulation shows an increasing negative bias with time because of an incorrect balance between methane sources and sinks. All inversions (posterior solutions) are successful in correcting this bias, including its seasonality.





**Figure 5. Optimized global distributions of 2013-2015 non-wetland methane emissions using GOSAT, AIRS, and GOSAT+AIRS observations. Prior emissions are shown in (a). The average posterior/prior ratios from 2013-2015 for inversions with each set of observations are shown in (b)-(d). Total emissions are inset in (a)-(d) with their error standard deviations. Averaging kernel sensitivities (diagonal elements of the averaging kernel matrix) averaged over 2013-2015 are shown in (e)-(g). The averaging kernel sensitivities represent the ability of the inversion to constrain the posterior solution independently from the prior estimate (1= fully, 0 = not at all). The degrees of freedom for signal (DOFS) for the 1009 4°x5° grid cells averaged over 3 years are inset.**

The inversions optimize both methane emissions and OH concentrations. Figure 5 shows the prior non-wetland emissions and 2013-2015 posterior/prior correction factors for all three inversions, as well as the averaging kernel sensitivities. The GOSAT-only inversion (Figure 5b) shows upward corrections to the southern United States, Brazil, and East Africa, and



downward corrections to East Asia and parts of Russia, consistent with Zhang et al. (2021) who used similar prior estimates. The AIRS-only inversion shows generally similar results but weaker averaging kernel sensitivities. Results from the AIRS-only inversion are consistent with those of the GOSAT-only inversion with the exception of strong upward corrections over Brazil, Argentina, and India, which together cause much higher global methane emissions in the AIRS-only solution than the two solutions constrained by GOSAT observations. The greater power of the GOSAT data to constrain emissions on the 4°x5° grid is measured by the DOFS (144 for GOSAT, 33 for AIRS). Adding AIRS observations to GOSAT increases the DOFS by only 4% as the information on emissions from these two sensors overlaps.

We find small (<10 Tg a$^{-1}$) changes from year to year for methane emissions in all solutions, and most of these changes are attributed to non-wetland emissions. This is consistent with the solutions in Yin et al. (2021) who find global methane emissions changes over 2013-2015 on the order of 1-2%.





**Figure 6. Monthly mean 2013-2015 wetland emissions for the 14 WetCHARTs subcontinental regions as defined by Bloom et al. (2017). Prior emission estimates from the mean of the WetCHARTs inventory ensemble are compared to posterior emissions from the GOSAT, AIRS, and GOSAT+AIRS inversions. The degrees of freedom (DOFS) for signal aggregated to 14 regions x 12 months = 168 state vector elements are also given.**

Figure 6 shows inversion results for the seasonality of wetland emissions in the 14 subcontinental regions of the WetCHARTs inventory used as prior estimate. The seasonality and magnitude of the GOSAT and GOSAT+AIRS posterior estimates are consistent with Zhang et al. (2021), who used a similar wetland state vector but with more years of GOSAT data. Our posterior produces negative emissions in Boreal North America in the spring, which are also seen in the solution of Zhang et al. (2021). They attribute these negative emissions to potential soil sinks in the region. Remarkably, the AIRS-only inversion shows the same feature. Remarkably, the posterior global sum of non-wetland and wetland emissions in the GOSAT and GOSAT+AIRS inversions is lower than the prior estimate, even though the prior simulation is biased low (Figure 4). This is because of a compensating decrease in $\overline{[\text{OH}]}$, as analyzed below.



### 4.2 Quantifying global mean OH concentrations independently of emissions

We now turn our attention to the ability of the satellite observations to constrain the global annual mean OH concentration, $[\overline{OH}]$, independently of emissions and for individual years. Let $E$ denote the global annual mean methane emission rate. The annual rate of change in atmospheric methane mass, $\Delta m/\Delta t$, is given by

$$\frac{\Delta m}{\Delta t} = E - k[\overline{OH}]m - L \tag{13}$$

where $k$ is the rate constant for oxidation of methane by tropospheric OH with a suitable temperature kernel (Prather and Spivakovsky, 1990) and $L$ is the sum of other minor sinks with $L \ll k[\overline{OH}]m$. Considering that $\Delta m/\Delta t$ is set by the observations used in the inversion, and $L$ is minor and not optimized, we see that corrections to $E$ and $[\overline{OH}]$ are necessarily correlated. In order to constrain $[\overline{OH}]$ we need independent information on emissions. The lower-atmosphere gradients over land observed by GOSAT can provide that information, as pointed out by Zhang et al. (2021) and shown in Section 4.1, but the AIRS TIR measurements cannot and this is reflected in the low DOFS of Figures 5 and 6. We focus therefore on the GOSAT and GOSAT+AIRS observing configurations to evaluate their capability to constrain $[\overline{OH}]$ in individual years separately from emissions.

Figure 7 shows the corrections to $E$ and $[\overline{OH}]$ for individual years from the inversions. The inversions apply a systematic correction to $[\overline{OH}]$ in all three years, reflecting bias in the prior [OH], and a smaller interannual variability. The right panels show the rows of the reduced averaging kernel matrix summing emissions globally (Eq. (8)) and diagnosing the ability of the inversion to correct separately $[\overline{OH}]$ and $E$ in individual years. We find that the averaging kernels for $[\overline{OH}]$ in individual years are strongly peaked, with no significant aliasing from emissions and only minor aliasing with $[\overline{OH}]$ for other years. We conclude that $[\overline{OH}]$ can be optimized for individual years and independently of emissions. Some aliasing of the inverse solution to $[\overline{OH}]$ across years is to be expected in view of the long lifetime of methane but we are still able to capture individual years and thus interannual variability of $[\overline{OH}]$. GOSAT+AIRS provides only slightly more information than GOSAT alone. A similar averaging kernel analysis by (Maasakkers et al., 2019) for 2010-2015 GOSAT observations found that the observations could constrain the average $[\overline{OH}]$ over all years but not the interannual variability. In that study the emission trend was imposed to be linear, which would strongly detract from the ability to independently constrain interannual variability of $[\overline{OH}]$.





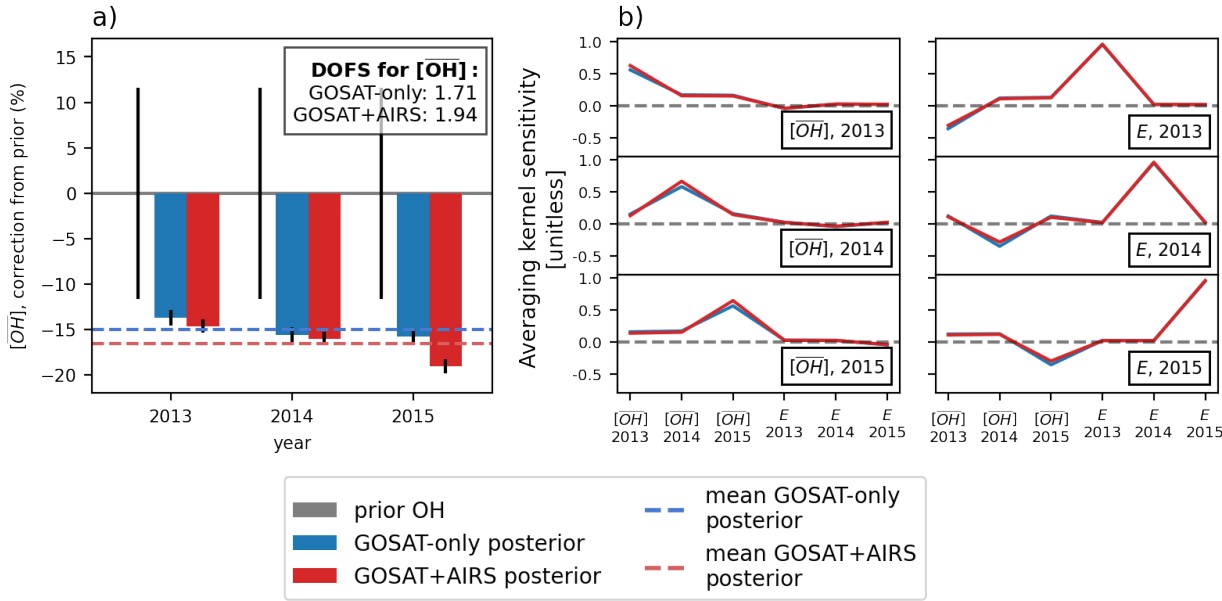

**Figure 7. Ability of inversions using GOSAT and GOSAT+AIRS methane observations to quantify global annual mean**
**tropospheric [OH] for individual years and independently from emissions. (a) 2013-2015 percentage corrections to the [OH] prior**
**estimate. Prior and posterior error standard deviations are shown as error bars. DOFS are shown inset (DOFS = 3 would imply**
**perfect separate quantification of [OH] in individual years). (b) Rows of the reduced averaging kernel matrix describing the ability**
**of the observing system to separately quantify emissions (*E*) and [OH] for the individual years. A perfect observing system would**
**have an averaging kernel sensitivity of 1 for the reduced state vector element of interest (perfect characterization) and 0 for other**
**elements (no sensitivity of the solution to other elements).**

## 4.3 Resolving spatial and seasonal patterns in OH concentrations

We now investigate the ability of the methane observations to constrain the spatial and seasonal variations of OH

concentrations. Figure 8 shows the corrections to OH concentrations from the inversion as a function of latitude, along with

430 the corresponding rows of the averaging kernel matrix. We find that GOSAT and GOSAT+AIRS provide only weak

constraints on the OH latitudinal distribution because prior errors from the ACCMIP ensemble are highly correlated (Figure

3). We are unable to resolve the midlatitudes, where averaging kernel rows show higher sensitivity to the adjacent tropical

latitude band, and almost no sensitivity to the midlatitudes themselves. There is some information on the interhemispheric

ratio of OH concentrations, with the inversion decreasing the NH/SH ratio from 1.11±0.08 in the prior estimate to 1.01±0.02

(for GOSAT) and 1.04±0.01 (for GOSAT+AIRS). This is consistent with previous inversions of methane observations

showing downward corrections in the NH/SH ratio (Zhang et al., 2021) and independent evidence from MCF observations

that current model NH/SH ratios are too high (Naik et al., 2013; Patra et al., 2014). Nevertheless, we see from the averaging

kernels that there is significant aliasing of the information between the northern and southern tropics, because errors are



highly correlated across models (Figure 3). It could be that the ensemble of ACCMIP models exaggerates the error

correlation on account of using the same anthropogenic emissions, but OH in the tropics is more sensitive to lightning, fires, and clouds which vary across the models.

The seasonal cycle for [OH] is shown in Figure 9. We find from the averaging kernel matrix that the inversion provides significant information on the seasonality of [OH] in the two hemispheres, despite the smearing across latitudinal bands

found in Figure 8. There is some aliasing between adjacent seasons but winter and summer are well separated, mainly for the tropics since there is little information from mid-latitudes (Figure 8). The GOSAT+AIRS inversion increases the amplitude of the seasonal cycle in both hemispheres. The posterior seasonal patterns from the GOSAT and GOSAT+AIRS inversions do not differ significantly from the prior, which demonstrates a good understanding of the OH seasonality on the hemispheric scale.





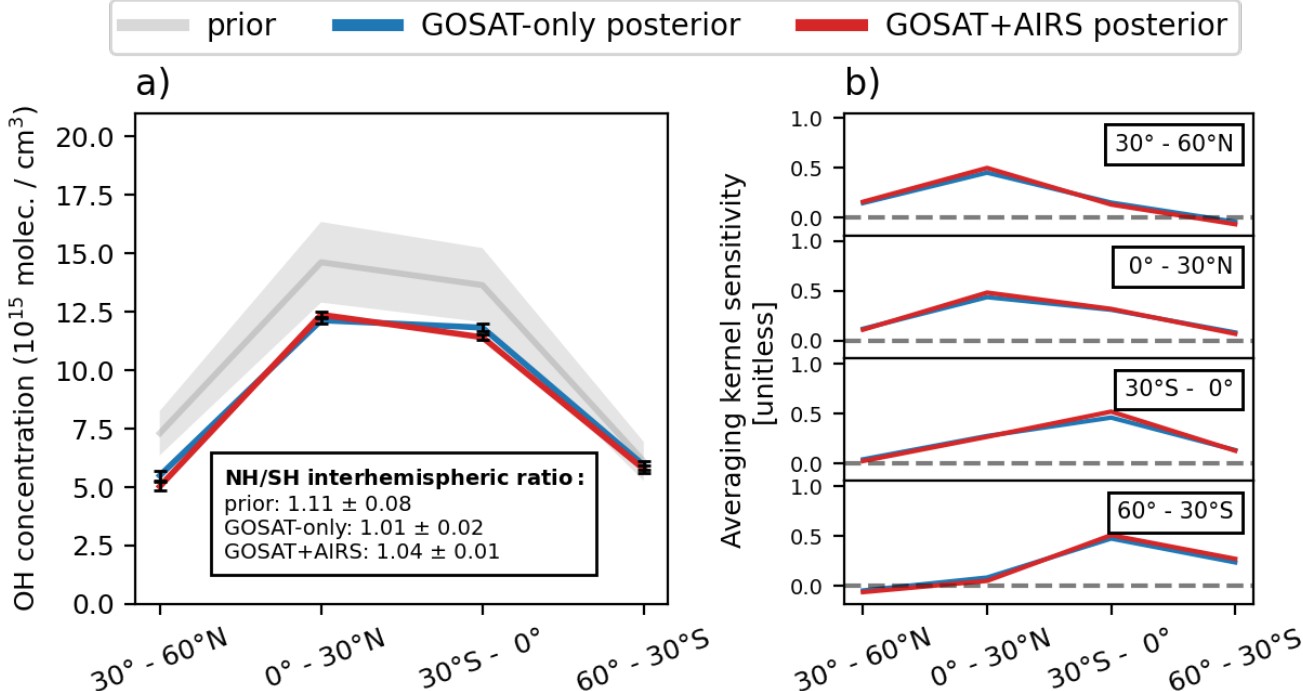

**Figure 8. Ability of inversions of GOSAT and GOSAT+AIRS methane observations to resolve the latitudinal variability of OH concentrations. (a) Latitudinal distribution of mass-weighted tropospheric [OH] in the prior estimate (prior error standard deviation in shading) and in the posterior estimates. The NH/SH interhemispheric ratio and its error standard deviation are inset. (b) Rows of the reduced averaging kernel matrix describing the ability of the observing system to separately quantify [OH] in different latitudinal bands. A perfect observing system would have an averaging kernel sensitivity of 1 for the reduced state vector element of interest (perfect characterization) and 0 for other elements (no error correlation).**

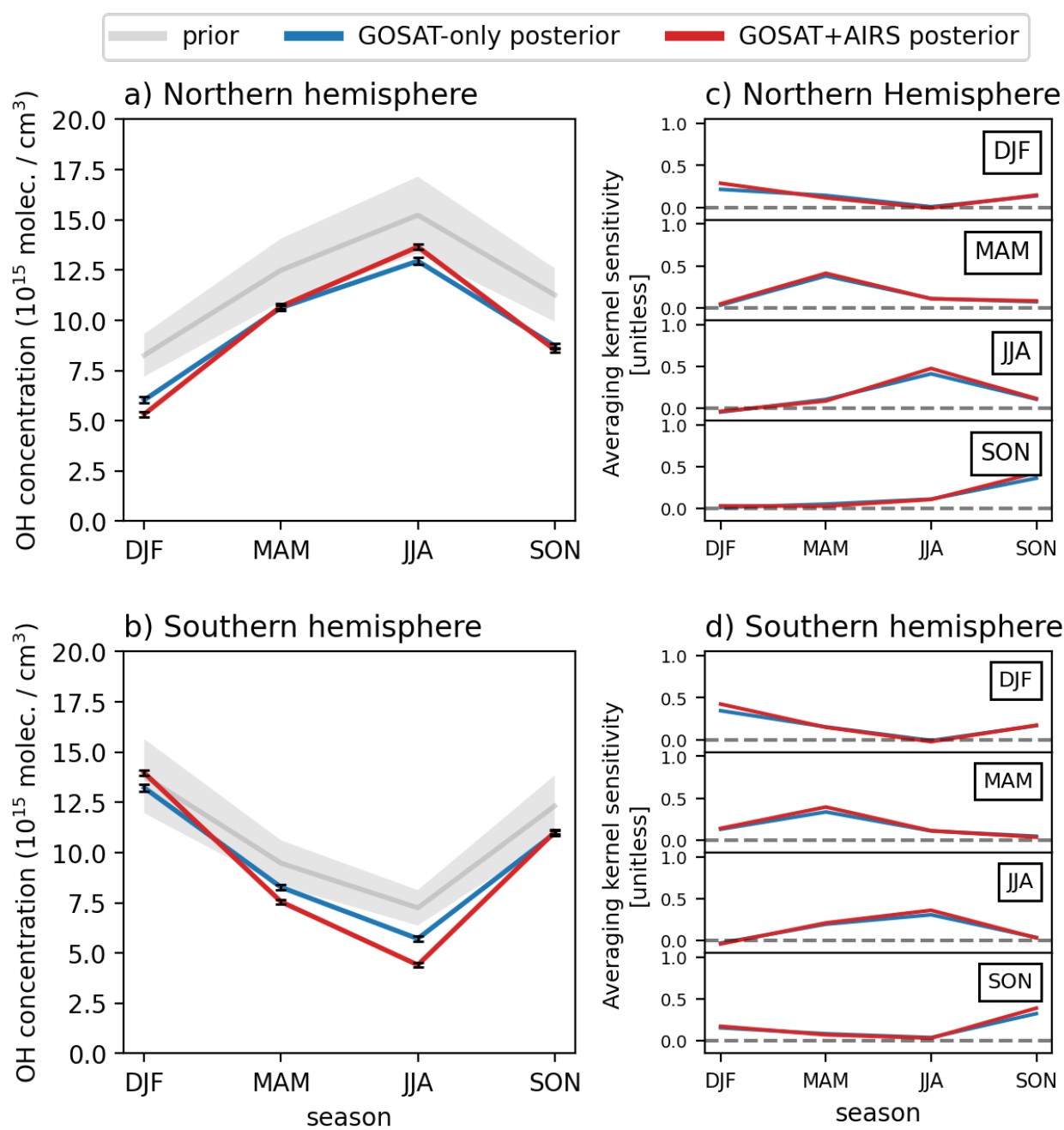

**Figure 9. Same as Figure 8 but for the seasonality of OH concentrations in each hemisphere.**




We have found that the ability of the inversion to optimize spatial and temporal features of the OH distribution is limited by prior error correlations from the independent knowledge expressed by the ACCMIP models. We examined the effect of these prior error correlations in sensitivity simulations for GOSAT-only inversions in which we either assumed no error correlations between OH state vector elements, or error correlations only for the interannual variability of $[\overline{OH}]$ as described by Eq. (10). Aggregated errors on $[\overline{OH}]$ were scaled to be the same in all inversions as described in Sect. 3. Fig. 10 shows the results for the GOSAT-only inversion. Constraints on $[\overline{OH}]$ are similar across all inversions, as would be expected since our base full-correlations inversion can effectively constrain that quantity for individual years. The inversions without error correlations show larger perturbations to the latitudinal distribution of [OH], with higher values at mid-latitudes and lower in the tropics, and a greater shift to the southern hemisphere. The spatial error correlations imposed by the ACCMIP models (Figure 3) suppress these changes in the base inversion. To the extent that the ACCMIP ensemble fairly represents error correlations on the OH distribution, ignoring that prior information results in overfit to observations. The seasonality in each hemisphere is better constrained by the observing system because there is more contrast between summer and winter, with northern and southern tropics being opposite in seasonal phase. However, we find that ignoring seasonal error correlations in the no-correlation and correlated-years inversions results in opposite corrections to OH concentrations in spring and summer of the northern hemisphere which are in fact highly correlated in the ACCMIP models (Figure 3).





**Figure 10. Sensitivity of [OH] inversion results to the prior error correlations imposed for interannual, seasonal, and latitudinal variability. Results are shown for the 2013-2015 GOSAT-only inversion, for our base inversion with full error correlations from the ACCMIP ensemble (same results as in Figures 7-9) and for inversions with no [OH] error correlations or with [OH] error correlations for individual years only. Panels show (a) annual mean $\overline{\text{OH}}$ for individual years, (b) 2013-2015 latitudinal distribution, and (c, d) 2013-2015 seasonal variations for the northern and southern hemispheres. Prior error standard deviations are shown as vertical bars and shading. The correlated-years and no-correlation inversions show the same latitudinal and seasonal variations of [OH].**



## 5 Conclusions

We examined the ability of satellite observations of atmospheric methane to quantify different features of the tropospheric OH distribution including global multi-year mean, interannual variability in the global mean, interhemispheric ratio, intra-hemispheric latitudinal variation, and seasonality. The work was motivated by the need to find a replacement proxy for tropospheric OH as methylcholoroform (MCF) concentrations fall below detectable levels, and to explore how much information can be extracted from the satellite observations.

We used for this purpose a 3-year (2013-2015) analytical inversion of GOSAT (SWIR) and AIRS (TIR) satellite observations. SWIR observations have near-unit sensitivity for the whole atmospheric column but are limited to daytime and (mainly) land. TIR observations are sensitive mainly to the middle/upper troposphere but include nighttime and oceans.

Several previous inversions investigated the ability of satellite observations of methane to quantify the OH distribution but did not properly account for prior error correlations in that distribution. Here we provide detailed accounting of this error correlation including for global mean OH and interannual variability using MCF, and for spatial and seasonal variations using the ACCMIP ensemble of 11 global atmospheric chemistry models. We find strong prior error correlations between latitude bands and seasons.

Optimizing OH concentrations from satellite observations of methane requires independent information on emissions and the SWIR observations are essential for that purpose. We find that a GOSAT-only inversion can effectively constrain global mean OH and its interannual variability independently of emissions, thus providing information comparable to MCF. Adding AIRS observations to the inversion does not significantly improve the constraint.

The ability of the inversion to resolve the latitudinal variability of OH is very limited because of strong error correlation across latitudes in the ACCMIP ensemble. Not accounting for this error correlation would result in overfit to observations. There is in particular no information on OH at mid-latitudes. The inversion provides some information on the interhemispheric OH ratio, and this is important for interpreting the corresponding gradient in methane observations (East et al., 2024). There is also some information on seasonality of OH concentrations, and the inversion confirms the prior seasonality from the ACCMIP models.

TROPOMI observations starting in May 2018 provide much denser SWIR data for methane than GOSAT (Lorente et al., 2021), allowing finer-grained quantification of emissions, but for coarse global-scale inversions as presented here the GOSAT observations offer similar information content as TROPOMI (Qu et al., 2021) and are of higher quality (Balasus et al., 2023). Beyond GOSAT and TROPOMI, the constellation of satellite observations of methane is rapidly expanding



(Jacob et al., 2022), providing an effective vehicle to monitor tropospheric OH and its interannual variability in the future. This will be important for interpreting future methane trends and for improving our understanding of the factors controlling

tropospheric OH. Improving the ability of the inversions to constrain the spatial variability of OH will require partnership with atmospheric chemistry models to resolve error correlations, possibly through observations of other trace gases such as CO.

## 6 Data Availability

The GOSAT methane retrievals version 9.0 are available at https://dx.doi.org/10.5285/18ef8247f52a4cb6a14013f8235cc1eb

(Parker and Boesch, 2020). The AIRS methane retrievals are available at
https://disc.gsfc.nasa.gov/datasets/TRPSDL2CH4AIRSFS_1/summary (Kulawik et al., 2021). Oil, gas, and coal emissions from the GFEIv1.0 inventory are available at
https://dataverse.harvard.edu/dataset.xhtml?persistentId=doi:10.7910/DVN/HH4EUM&version=1.0 (Scarpelli et al., 2020).
Methane emissions from EDGAR v4.3.2  are available at https://edgar.jrc.ec.europa.eu/dataset_ghg432 (Crippa et al., 2018).

Wetland emissions from WetCHARTs v1.0 are available at https://doi.org/10.3334/ORNLDAAC/1502 (Bloom et al., 2017). The OH fields from the ACCMIP ensemble of models is available at
https://catalogue.ceda.ac.uk/uuid/ded523bf23d59910e5d73f1703a2d540 (Shindell et al., 2011).

## 7 Author Contributions

EP, DJJ, and JW contributed to the study conceptualization. JW provided the AIRS data. EP conducted the data and

modeling analysis with contributions from DJJ, ZC, JE, MPS, LB, JDM, HN, ZQ, YZ, and JW. EP and DJJ wrote the paper with contributions from all authors.

## 8 Competing Interests

The contact author has declared that none of the authors has any competing interests.

## 9 Acknowledgments

This work was funded by the NASA Carbon Monitoring System (CMS) and the NOAA AC4 program. This material is based upon work supported by the National Science Foundation Graduate Research Fellowship under Grant No. (DGE1745303). This work was funded in part by an appointment to the NASA Postdoctoral Program at the Jet Propulsion Laboratory, California Institute of Technology, administered by Oak Ridge Associated Universities under contract with NASA. Part of





this research was carried out at the Jet Propulsion Laboratory, California Institute of Technology, under a contract with the

National Aeronautics and Space Administration. Y. Zhang was supported by NSFC (42275112).



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
