# Peer review of "What can we learn about tropospheric OH from satellite observations of methane?"

_EGUsphere, 2024_

## Referee Comment (RC2)

Review of *"What can we learn about tropospheric OH from satellite observations of methane?"* by Penn et al.

My sincere apologies for the delay in my review.

Penn et al. present global inversions of methane and OH using GOSAT, AIRS, and GOSAT+AIRS from 2013-205. They find that both AIRS-only and GOSAT-only inversions have similar performance. They argue that GOSAT+AIRS-inversion is not substantially different than that GOSAT-only inversion and later argue that AIRS does not add much value to the inversion. The final argument is that GOSAT can independently constrain some aspects of methane and OH. Spatial patterns of OH do not seem well constrained. The paper is generally well written and the figures are quite clear. I have some comments below, but most of them are rather minor.

In this reviewers opinion, the main issue that needs to be addressed is the language regarding the performance of the different satellites. Previous work (including papers from some of the same authors) have argued that the combination of SWIR and TIR observations are quite valuable. This paper seems to find the opposite. This reviewer feels that the authors should clarify some of that language as the value of SWIR+TIR is being called into question. This can be easily addressed by adding just a bit of text to explicitly state where this SWIR+TIR is, or is not, valuable. This reviewer recommends minor revisions.

**Comments**

**1.) Previous work using SWIR and TIR**
The authors mention studies that developed combined products using SWIR and TIR in the past (Worden, Schneider, Kuze, and Suto). These seem to be most relevant, yet the findings from those papers are only briefly mentioned around Line 95. This is in contrast to other work that is discussed in detail in the preceeding paragraphs. Do these other papers using similar data and methods reach similar conclusions?

**2.) Choice of state vector**
How do the authors decide on the state vector? They separate the state vector into wetland emissions, non-wetland emissions, and OH. Is there sufficient sectoral information to attribute fluxes to wetland and non-wetland? It does not seem like they use any observations that would allow for that sort of separation.

**3.) Time period**
Why do the authors limit the study to 2013-2015? AIRS and GOSAT both have data extending much longer. Their Jacobian is quite small once constructed and the construction can be done in parallel. 3 years is very short given the interannual variability in some of their sources and sinks.

**4.) Prior error covariance for OH**
How do the authors decide on the off-diagonal elements in their error covariance matrix for OH? It seems like the 0.1 corresponds to the 10% systematic error. But that

systematic error would not be the off-diagonal term in the error covariance matrix. That off-diagonal term should signify how OH will co-vary across years. This reviewer would expect that OH in 2015 would co-vary more with 2014 than 2013, but that is not what their matrix indicates.

**5.) Off-diagonal elements in the averaging kernel**
Why don't the authors include the AIRS-only inversion in Figure 7? It would be interesting to see how AIRS alone performs. The authors argue that AIRS cannot resolve methane. Showing the low performance for AIRS would give the readers confidence that the other Averaging Kernels are meaningful.

Same for Figures 8 and 9.

My recollection was that Worden et al (2015) showed that there was value in having the combination of GOSAT and TES. Why does AIRS not provide any value in this study?

My main question at the end of the manuscript was about the relative value of each set of satellite data. The authors show that AIRS-only inversion performs similarly to the GOSAT-only inversion (Figs 4-6). But the authors later argue that AIRS does not add anything beyond GOSAT. I think this needs to be more clearly explained. The only real discussion I saw of this was on Page 16. I think this discussion needs to be expanded and laid out more clearly.

**Specific Comments**

**Ordering of references:** it seems the authors have ordered the citations alphabetically. This seems atypical. I would usually expect the first paper to show something to be the first paper cited. As an example, Lines 38-39 when discussing OH. This is pervasive through the manuscript.

**Lines 52-53:** I think this should say "lifetime of CH4". I don't think the authors mean to say the lifetime of OH is 10 years.

**Lines 159-160:** How do the authors rule out a bias in GEOS-Chem's free tropospheric methane?

**Line 263:** Missing a period before "We determine…".

**Figure 3:** Figure 3 is great.

**Figure 4:** Add AIRS-only to the left panel and GOSAT-only to the right panel. It would show how an AIRS-inversion performs against GOSAT and vice versa.

**Figure 5:** Emissions in panel d are less than b and c. Using both GOSAT and AIRS results in a decrease from the prior. Why?

**Lines 386-387:** Two sentences in a row start with "Remarkably".

---

## Author Comment (AC1)

We thank the reviewers for their suggestions and comments on the manuscript. Below, we have replied to each review and have detailed the corresponding edits that we have made to the manuscript. We have listed the reviewer comments in *black italic* and the replies in blue.

**Reviewer #1**

*In Penn et al, the authors perform an inversion using GEOS-Chem and methane from GOSAT and AIRS to optimize both methane emissions and OH concentrations. Including both GOSAT and AIRS allows the authors to determine the effect of including methane from both shortwave (GOSAT) and thermal (AIRS) infrared instruments. They find that including AIRS in the inversion provides little additional information beyond that found from using GOSAT alone. While the method cannot resolve information on OH at mid-latitudes due to correlation of errors, it was able to provide information on interannual variability and the interhemispheric ratio. This was an interesting paper that both demonstrates the need for continued methane observations by satellites as well as their utility in constraining OH at the global and hemispheric scales. It is suitable for publication in ACP once the very minor comments listed below are addressed.*

Minor Comments:

*Line 49: Should be "resulting in a lifetime of ~1 second". Currently missing "of".*

We have corrected as suggested.

*Line 56: Also differences in UV flux in the troposphere. See, for example, Nicely et al, 2020.*

We have added this as suggested, and included the reference.

*Line 159: You use a uniform correction of 19 ppb but figure 1 shows a clear latitudinal bias in AIRS with respect to the GEOS-Chem simulation you are using as "truth". Why not use a latitudinally dependent correction? In the end, I imagine this doesn't really affect your results since including AIRS doesn't seem to have much benefit.*

The rationale for a latitudinal correction is not clear and it could be criticized for removing information. We now say so in the text on line 161:

"Figure 1 shows additional latitudinal differences between AIRS and GOSAT but these may provide information for the inversion and we have no rationale to remove them."

*Line 176: Where do the 50% and 20% numbers come from? Are they just "best guesses", in which case, how sensitive are your results to these numbers?*

Thank you for the comment. We have clarified this on line 181:

"This calculation is insensitive to the magnitudes of the perturbations because the forward model is strictly linear in the relationship of concentrations to emissions, and the assumption of linearity is also acceptable for the relationship to OH concentrations in a 3-year simulation."

*Line 211: Similar question here as for line 176. What's your justification for using an error of 50%?*

We have clarified in the text on line 216:

"Prior error standard deviations for non-wetland emissions are assumed to be 50% of emissions for each 4°x5° grid cell with no error covariance between grid cells, as in previous studies (Maasakkers et al., 2019; Zhang et al., 2021). The effect of this prior error is reflected in the averaging kernel sensitivities."

*Line 385: You refer to "Boreal North America" as "East Canada" in Figure 6. Be consistent with labeling.*

We have corrected as suggested.

*Line 387: Probably shouldn't start two consecutive sentences with "remarkably".*

We have corrected as suggested.

*Conclusions: One of the main conclusions from this work seems to be that including information from AIRS doesn't really seem to improve the inversions and that information from the SWIR retrieval (GOSAT, in this case) is sufficient. It could be helpful to have a discussion on whether this is just a result of limitations of the AIRS instrument, or if you think other TIR instruments could prove more useful. What would need to be done to make the TIR retrievals more useful for these types of studies, either improvements to the retrievals or to the instruments themselves? Similarly, when discussing the inability of this methodology to resolve information on OH across more latitude bands, what could be done to improve this, if anything? Using a different or larger set of independent simulations? Or is this simply a limitation of this methodology and it is unlikely to ever yield spatially resolved information on OH distributions?*

We have discussed this on line 535:

"Acquiring finer regional-scale information on OH is of great interest but the long lifetime of methane likely limits the information that it can provide to the global scale, even with improved satellite instruments. Satellite observations of shorter-lived species driving OH chemistry including $H_2O$, $O_3$, $CO$, $NO_2$, and HCHO provide fine-scale information on OH through chemical data assimilation (Miyazaki et al, 2020), but the results may be biased by errors in the chemical mechanisms (Travis et al., 2020; Shah et al., 2023). The global-scale information on OH concentrations available from methane observations can be used for independent evaluation of such data assimilation products."

*References:*

*Nicely, J. M., Duncan, B. N., Hanisco, T. F., Wolfe, G. M., Salawitch, R. J., Deushi, M., et al. (2020). A machine learning examination of hydroxyl radical differences among model simulations for CCMI-1. Atmospheric Chemistry and Physics, 20(3), 1341-1361.*

We have included this reference.

**Reviewer #2**

*Review of "What can we learn about tropospheric OH from satellite observations of methane?" by Penn et al. My sincere apologies for the delay in my review. Penn et al. present global inversions of methane and OH using GOSAT, AIRS, and GOSAT+AIRS from 2013-205. They find that both AIRS-only and GOSAT-only inversions have similar performance. They argue that GOSAT+AIRS-inversion is not substantially different than that GOSAT-only inversion and later argue that AIRS does not add much value to the inversion. The final argument is that GOSAT can independently constrain some aspects of methane and OH. Spatial patterns of OH do not seem well constrained. The paper is generally well written and the figures are quite clear. I have some comments below, but most of them are rather minor. In this reviewers opinion, the main issue that needs to be addressed is the language regarding the performance of the different satellites. Previous work (including papers from some of the same authors) have argued that the combination of SWIR and TIR observations are quite valuable. This paper seems to find the opposite. This reviewer feels that the authors should clarify some of that language as the value of SWIR+TIR is being called into question. This can be easily addressed by adding just a bit of text to explicitly state where this SWIR+TIR is, or is not, valuable. This reviewer recommends minor revisions.*

***Comments***

***1.) Previous work using SWIR and TIR***
*The authors mention studies that developed combined products using SWIR and TIR in the past (Worden, Schneider, Kuze, and Suto). These seem to be most relevant, yet the findings from those papers are only briefly mentioned around Line 95. This is in contrast to other work that is discussed in detail in the preceeding paragraphs. Do these other papers using similar data and methods reach similar conclusions?*

We have added the following text on line 521:

"Retrievals combining SWIR and TIR information from the same instrument, such as GOSAT-2 (Kuze et al., 2022; Suto, 2022), could possibly improve the constraint by separating lower and upper tropospheric contributions to the methane column.  This would need to be examined in future work."

***2.) Choice of state vector***
*How do the authors decide on the state vector? They separate the state vector into wetland emissions, non-wetland emissions, and OH. Is there sufficient sectoral information to attribute fluxes to wetland and non-wetland? It does not seem like they use any observations that would allow for that sort of separation.*

We have added the following text on line 172:

"Separate characterization of wetland and non-wetland emissions is done on the basis of assumed subcontinental spatial coherence and seasonality of the prior wetland emission estimates (Maasakkers et al., 2019; Zhang et al., 2021)."

**3.) Time period**
*Why do the authors limit the study to 2013-2015? AIRS and GOSAT both have data extending much longer. Their Jacobian is quite small once constructed and the construction can be done in parallel. 3 years is very short given the interannual variability in some of their sources and sinks.*

We have added an explanation on line 524:

"We conducted the inversion for only three years (2013-2015) to demonstrate the capability for constraining OH interannual variability. Qu et al. (2024) recently conducted an inversion of the full GOSAT record from 2011 to 2022 to quantify the OH interannual variability over that 13-year period."

**4.) Prior error covariance for OH**
*How do the authors decide on the off-diagonal elements in their error covariance matrix for OH? It seems like the 0.1 corresponds to the 10% systematic error. But that systematic error would not be the off-diagonal term in the error covariance matrix. That off-diagonal term should signify how OH will co-vary across years. This reviewer would expect that OH in 2015 would co-vary more with 2014 than 2013, but that is not what their matrix indicates.*

We have added the following text to clarify the construction of the OH error covariance (line 291):

"…the off-diagonal terms enforce the assumption of a 10% systematic error (perfectly correlated across all years). The OH interannual variability is assumed not to be correlated across years."

We agree that it could be reasonable for adjacent years for OH to have higher correlation. However, we did not have ACCMIP multi-model runs for multiple years available to determine appropriate correlations.

**5.) Off-diagonal elements in the averaging kernel**
*Why don't the authors include the AIRS-only inversion in Figure 7? It would be interesting to see how AIRS alone performs. The authors argue that AIRS cannot resolve methane. Showing the low performance for AIRS would give the readers confidence that the other Averaging Kernels are meaningful.*

*Same for Figures 8 and 9.*

*My recollection was that Worden et al (2015) showed that there was value in having the combination of GOSAT and TES. Why does AIRS not provide any value in this study?*

*My main question at the end of the manuscript was about the relative value of each set of satellite data. The authors show that AIRS-only inversion performs similarly to the GOSAT-only inversion (Figs 4-6). But the authors later argue that AIRS does not add anything beyond GOSAT. I think this needs to be more clearly explained. The only real discussion I saw of this was on Page 16. I think this discussion needs to be expanded and laid out more clearly.*

Thank you for your comment, we agree that we could add more detail on the AIRS-only inversion, and have addressed it in several parts of the paper, outlined below:

We have added the AIRS-only inversion results and averaging kernels to Figures 7, 8, and 9.

We have added discussion on how we infer that AIRS does not provide additional value to GOSAT on line 374:

"Adding AIRS observations to GOSAT increases the DOFS by only 4%, indicating that the information on emissions from these two sensors has extensive overlap."

And the following text on line 415 to explain excessive OH in the AIRS-only inversion:

"The AIRS-only inversion has excessive [$\overline{\text{OH}}$] to offset its poorly constrained and excessive global emission (Figure 5)."

And the following text on line 521 to address the potential value of TIR observations:

"Retrievals combining SWIR and TIR information from the same instrument, such as GOSAT-2 (Kuze et al., 2022; Suto, 2022), could possibly improve the constraint by separating lower and upper tropospheric contributions to the methane column. This would need to be examined in future work."

**Specific Comments**

*Ordering of references: it seems the authors have ordered the citations alphabetically. This seems atypical. I would usually expect the first paper to show something to be the first paper cited. As an example, Lines 38-39 when discussing OH. This is pervasive through the manuscript.*

We have corrected as suggested.

*Lines 52-53: I think this should say "lifetime of CH4". I don't think the authors mean to say the lifetime of OH is 10 years.*

We have corrected as suggested.

*Lines 159-160: How do the authors rule out a bias in GEOS-Chem's free tropospheric methane?*

We have added a discussion on line 159:

"Although errors in the GEOS-Chem vertical profiles of methane mixing ratios would affect this intercomparison platform, we see in Figure 1 that the AIRS bias extends over background regions where the vertical profile would be uniform."

*Line 263: Missing a period before "We determine…".*

We have corrected as suggested.

*Figure 3: Figure 3 is great.*

Thank you.

*Figure 4: Add AIRS-only to the left panel and GOSAT-only to the right panel. It would show how an AIRS-inversion performs against GOSAT and vice versa.*

We have corrected as suggested.

*Figure 5: Emissions in panel d are less than b and c. Using both GOSAT and AIRS results in a decrease from the prior. Why?*

We have explained this on line 375:

"The GOSAT+AIRS inversion results largely follow those of the GOSAT-only inversion but the global posterior emission estimate is lower than in either the GOSAT-only or AIRS-only inversions because of selected regions where AIRS has influence, such as to decrease emissions in China."

*Lines 386-387: Two sentences in a row start with "Remarkably".*

We have corrected as suggested.

---

## Author Response (AR2)

We thank the reviewer for their suggestions and comments on the manuscript. Below, we have replied to the review and have detailed the corresponding edits that we have made to the manuscript. We have listed the reviewer comments in *black italic* and the replies in blue.

*I don't feel that the authors have addressed my minor comments from the first round. I suggest Minor Revisions again.*

*The main point of this paper is to tell us what we can learn about OH from satellite observations of methane. That is the question posed in the title of the paper. Previous work, including from many of these same authors, have looked at a similar question but reached very different conclusions. For example Worden et al. (2015) and Zhang et al. (2018) both found benefits from including TIR measurements. The OSSE from Zhang et al. (2018) found TIR measurements to be better at separating OH changes from methane emissions than SWIR measurements (their Figs 7-8). Worden et al. (2015) found that combining SWIR and TIR measurements to be helpful for assessing lower tropospheric methane. The findings from Penn et al. seem counter to previous work from these same authors. As a reader I am left wondering why.*

*The justification in the updated text seems to indicate that they think using SWIR and TIR from the same instrument might help, but its not clear why. The DOFS likely wouldn't change much from what was considered here. Why does would that give them more DOFS than AIRS did?*

We now better place our results in the context of Worden et al. (2015) and Zhang et al. (2018), and explain why SWIR and TIR measurements from the same instrument may do better. Specifically we have added the following text:

In Section 4.1:

"Our finding that AIRS does not add much information for optimizing methane emissions beyond GOSAT alone is not inconsistent with a previous finding by Worden et al. (2015) that TIR information from the TES satellite instrument improves the retrieval of lower tropospheric methane compared to a GOSAT-only retrieval. In our inversion, the GEOS-Chem forward model effectively provides the information to separate lower tropospheric methane from higher altitudes. An implication is that TIR observations are not necessary for enforcing that separation beyond the information from GEOS-Chem. "

In Section 4.2:

"Our finding that AIRS provides little information on $[\overline{OH}]$ beyond that provided by GOSAT contrasts with the Zhang et al. (2018) OSSE that found TIR methane observations to add significant information on emissions and $[\overline{OH}]$ relative to SWIR alone. That OSSE may have found a greater benefit from TIR because they assumed the SWIR and TIR synthetic observations to be perfectly consistent, while there are likely inconsistencies between the GOSAT and AIRS observations beyond our global correction (Figure 1) that translate into the differences between GOSAT-only and AIRS-only inversion results. Zhang et al. (2018) also gave the same weight to SWIR and TIR observations whereas we find that the weight for AIRS observations should be half of that for GOSAT based on optimization of the $\gamma$ coefficients

(Section 2.5). Beyond this, comparison of our results with Zhang et al. (2018) is difficult because they emulated different satellite instruments (TROPOMI for SWIR, CrIS for TIR) and did not report their assumed observational error variances."

and in Conclusions:

"Retrievals combining SWIR and TIR information from the same instrument, such as GOSAT-2 (Kuze et al., 2022; Suto, 2022), could possibly improve the constraint by separating lower and upper tropospheric contributions to the methane column being internally consistent."